# Barcode-free hit discovery from massive libraries enabled by automated small molecule structure annotation

Edith van der Nol[1,2,6], Nils Alexander Haupt [3,6], Qing Qing Gao[1], Benthe A. M. Smit [1], Martin Andre Hoffmann[4], Martin Engler-Lukajewski [4], Marcus Ludwig [4], Sean McKenna [1,2], J. Miguel Mata [1,2], Olivier J. M. Béquignon[1], Gerard van Westen [1], Tiemen J. Wendel [2,5], Sylvie M. Noordermeer [2,5], Sebastian Böcker [3] ✉ & Sebastian Pomplun [1,2] ✉

Affinity-selection platforms are powerful tools in early drug discovery, but current technologies – most notably DNA-encoded libraries (DELs) – are limited by synthesis complexity and incompatibility with nucleic acid-binding targets. We present a barcode-free self-encoded library (SEL) platform that enables direct screening of over half a million small molecules in a single experiment. SELs combine tandem mass spectrometry with custom software for automated structure annotation, eliminating the need for external tags for the identification of screening hits. We develop efficient, high-diversity synthesis protocols for a broad range of chemical scaffolds and benchmark the platform in affinity selections against carbonic anhydrase IX, identifying multiple nanomolar binders. We further apply SELs to flap endonuclease 1 (FEN1) – a disease related DNA-processing enzyme inaccessible to DELs – and discover potent inhibitors. Taken together, screening barcode-free libraries of this scale all at once represents an important development, enables access to novel target classes, and promises substantial impact on both academic and industrial early drug discovery.

The discovery of high-affinity ligands is crucial for virtually any drug discovery campaign. The pharmaceutical industry relies on vast collections of individual compounds (typically 0.5 to 4 million) and high-throughput screening (HTS) facilities in order to identify novel ligands for drug targets[1]. HTS has delivered numerous starting points for therapeutic compounds. Unfortunately, these libraries may cost billions of US dollars, and the required infrastructure is large and complex, limiting the availability of such massive platforms mainly to big pharma and a few academic settings.

Affinity selection technology enables the screening of large libraries in a single experiment[2–6]. These libraries are typically panned against immobilized target proteins, allowing for the separation of binders from non-binders. A crucial step in this process is the decoding of each hit, which is most commonly achieved using DNA or RNA barcodes attached to each ligand. Display technologies, such as phage display or mRNA display, utilize the natural translation machinery to convert genetic code into peptidic compounds, linked to their encoding oligonucleotides[7–9]. DNA-encoded libraries (DELs) feature small molecules linked to unique DNA sequences, enabling the exploration of a drug-like chemical space in the affinity selection setting[10–17].

While the DNA barcode is an essential component for hit decoding, it also represents the primary limitation in DEL technology in

[1]LACDR, Leiden University, Leiden, The Netherlands. [2]Oncode Institute, Utrecht, The Netherlands. [3]Chair for Bioinformatics, Institute for Computer Science, Friedrich Schiller University Jena, Jena, Germany. [4]Bright Giant GmbH, Hans-Knöll-Straße 6, Jena, Germany. [5]Department of Human Genetics, Leiden University Medical Center, Leiden, The Netherlands. [6]These authors contributed equally: Edith van der Nol, Nils Alexander Haupt. ✉e-mail: sebastian.boecker@uni-jena.de; s.j.pomplun@lacdr.leidenuniv.nl

terms of information stability and synthesis complexity. During library preparation, chemical reactions must be alternated with enzymatic ligation steps, and all transformations need to be both water- and DNA-compatible[18]. Many standard chemical reactions involve conditions that degrade DNA, and could compromise the chemical barcode[19]. Suitable, compatible reaction conditions have to be optimized and validated[20], further complicating the synthesis. Furthermore, the DNA tag is typically more than 50 times larger than the small molecule, which can potentially affect the selection process by restricting the binding pose diversity of each library member, or by interacting with the target and leading to false negatives or false positives[21]. This limitation becomes particularly problematic when the target protein has nucleic acid binding sites, making the screening for ligands of crucial drug targets like transcription factors or RNA-binding proteins very challenging[22].

Consequently, barcode-free selection is a highly desirable technology. However, current approaches that rely on mass spectrometry (MS) to identify selected compounds from tag-free libraries can process at most a few thousand compounds per sample[4,23–26]. Larger library sizes have been achieved only for peptidic compounds, which are structurally highly restricted and, compared to general small molecules, have unfavorable drug-like parameters[2,3,27–31].

Here, we report an affinity selection platform that screens barcode-free small molecule libraries with $10^4$ to $10^6$ members in a single run (Fig. 1). The approach features the combinatorial synthesis of drug-like compounds on solid phase beads, allowing for a wide range of chemical transformations and circumventing the complexity and limitation of DEL preparation. Compounds are annotated using their tandem MS fragmentation spectra, obviating the need for barcoding tags and, at the same time, enabling the distinction of hundreds of isobaric compounds. Recent progress in mass spectrometry instrumentation and computational methods for small molecule annotation are crucial factors for our platform. We show the feasibility of our decoding strategy on a diverse set of chemical scaffolds, prepared by a variety of chemical transformations, including cross-couplings, heterocyclizations, amide formation, nucleophilic aromatic substitution, and more. We then performed de novo discovery selections, panning libraries up to 750 k members against the two oncology targets carbonic anhydrase IX and the flap endonuclease-1 (FEN1),

resulting in the identification of nanomolar binders and inhibitors for both targets. Notably, FEN1 is a DNA-processing enzyme and therefore not suited for DEL selections. Overall, our SEL platform presents ideal features in terms of straightforward library preparation, information stability and unbiased screening capabilities.

## Results

### Library design and synthesis

Aiming at affinity selection with large and diverse self-encoded libraries, we established solid-phase synthesis protocols for the preparation of combinatorial libraries with different scaffold designs. By exploring different scaffolds, we aimed both at increasing diversity and at investigating the amenability of different molecular architectures for MS/MS-based decoding. In order to obtain high-quality combinatorial libraries, each reaction step needs to be efficient and high yielding. Self-encoded library 1 (SEL 1) is formed by the sequential attachment of two amino acid building blocks, followed by the addition of a carboxylic acid decorator using reaction conditions optimized for Fmoc-based solid phase peptide synthesis (Fig. 2a). Self-encoded library 2 (SEL 2) is based on a benzimidazole core decorated on three different positions. Based on previously described methodologies[32–34] and following systematic optimization, we established an efficient route towards trifunctional benzimidazoles (Fig. 2b, Supplementary Fig. 1). The benzimidazole decorators, which confer diversity to the library, are based on an amino acid building block, a primary amine and an aldehyde. We tested the scope of the nucleophilic aromatic substitution with a set of 92 primary amines, of which a large fraction resulted in reasonable conversion for combinatorial synthesis (> 65%) (Fig. 2b, Supplementary Fig. 2). We then investigated the heterocyclization efficiency using 95 aldehydes, with 65 resulting in a > 55% conversion to the final trifunctional compounds (Fig. 2b, Supplementary Fig. 3). Self-encoded library 3 (SEL 3) results from an amino acid building block linked to an aryl bromide and subsequently cross-coupled to a boronic acid, utilizing the palladium catalyzed Suzuki-Miyaura reaction[35]. We optimized reaction conditions on a selected scaffold (Supplementary Fig. 4) and then tested the scope of 19 bifunctional aryl bromides, of which 9 resulted in a > 65% conversion (Fig. 2c, Supplementary Fig. 5). Out of 86 boronic acids, 50 resulted in a > 65% conversion (Fig. 2c, Supplementary Fig. 6).

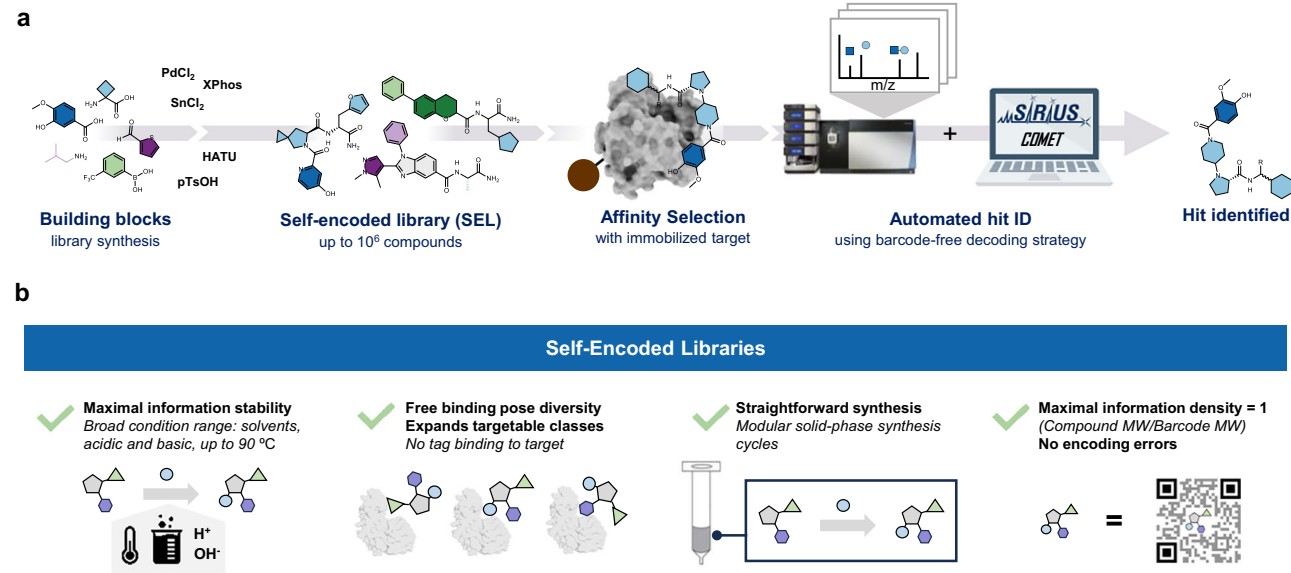

**Fig. 1 | Hit discovery with SELs. a** SEL workflow: large libraries with diverse scaffolds are prepared via solid phase split and pool synthesis, with a wide compatibility range of chemical transformation. The libraries are then released from beads and screened in solution against immobilized targets. Binders are enriched from the

rest of the library and analyzed via nanoLC-MS/MS. Hit compounds are identified and decoded by the custom-made COMET (COmbinatorial Mass Encoding decoding Tool) software. **b** Summary of SEL features, making it an excellent platform for rapid and straightforward early drug discovery.

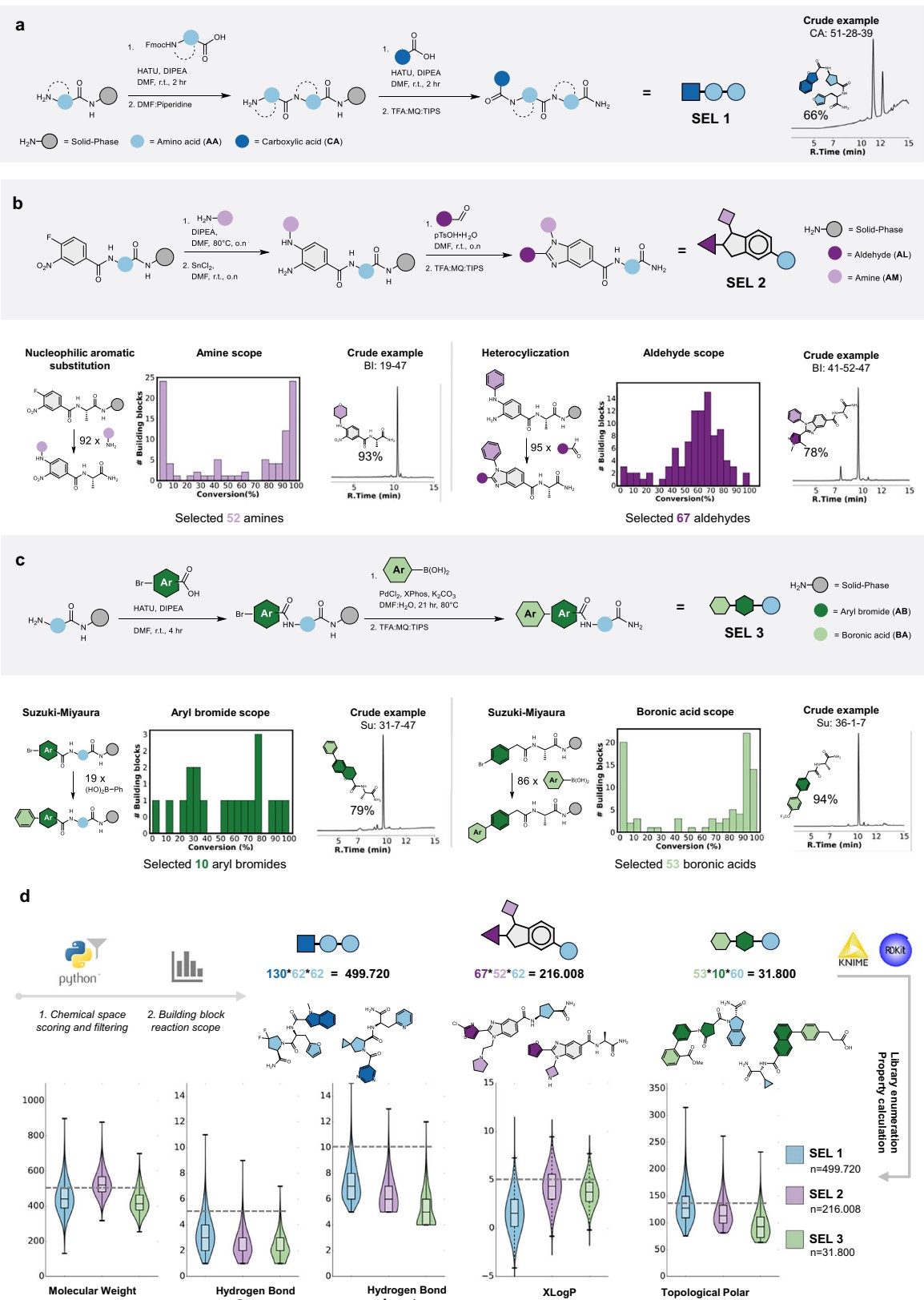

Representative crude LC-MS traces show the quality of the synthesis for each library scaffold (Fig. 2a–c).

Using a virtual library scoring script, we selected building blocks to generate libraries with optimized drug-like properties. For SEL 1, we anticipated no synthetic limitations, so we decided to select our initial set of building blocks based on their drug-like properties, while limiting isobaric fragments. We filtered a comprehensive building block (BB) catalog (containing 1681 Fmoc-amino acids and 6357 carboxylic acids from Chemspace) by availability and price, narrowing it down to 1000 BBs per position. With these, we enumerated a virtual library following the design of SEL1 with a billion members (1000*1000*1000). Each member was scored based on five Lipinski

**Fig. 2 | Library design and synthesis. a** SEL 1 is prepared from two amino acid building blocks decorated with a carboxylic acid. Standard solid-phase amide bond formation protocols result in products with excellent crude purity. **b** SEL 2, resulting in a benzimidazole scaffold with three decorators, is synthesized in a 7-step procedure. Optimization of all steps and reaction scope for the nucleophilic aromatic substitution and the heterocyclization are reported in detail in Supplementary Figs. 1–3 and summarized in the histograms. **c** SEL 3 is prepared from an amino acid, and a Suzuki-Miyaura reaction between an aryl bromide and a boronic acid. Cross-coupling optimization and reaction scope for the aryl bromides and boronic acids are reported in detail in Supplementary Figs. 4–6 and summarized in

the histograms. **d** Three high-diversity libraries were prepared from the selected building blocks. For SEL 1, 62 amino acids and 130 carboxylic acids were chosen based on favorable drug-like properties. SEL 2 and SEL 3 used building blocks, which in the reaction scope resulted in a crude product purity > 55%. Selected examples from each library emphasize the chemical diversity of the generated structures. The violin plots show the compliance of our libraries with several parameters important for drug development. Boxplots within the violin plots show the minimum, maximum, median, 75th percentile, and 25th percentile. Source data are provided as a Source Data file.

parameters: molecular weight (MW), logP, hydrogen bond donors (HBD), hydrogen bond acceptors (HBA), and topological polar surface area (TPSA)[36]. Each library member received a point for each satisfied parameter, which was then translated to a combined score per building block. This ranking allowed us to select and purchase top-scoring building blocks (62 amino acids and 130 carboxylic acids). Through solid phase split and pool synthesis, we generated SEL 1 with 499,720 members. Compared to the original enumerated library, all Lipinski parameters of our SEL 1 were substantially improved (Supplementary Fig. 7). For SEL 2 and 3, we used one of the optimized amino acids for BB1 and selected BBs with yields greater than 55% in scope analysis for the other positions (resulting in 216,008 members for the benzimidazole scaffold and 31,800 members for the Suzuki based library). Overall, the majority of all compounds in the three libraries satisfy the requirements for drug-like properties (Fig. 2d).

## Library decoding

A crucial step in the affinity selection workflow is the accurate identification of hit compounds. The final sample from an affinity selection process is always of unknown complexity and may contain up to a few hundred compounds. With the high degree of mass degeneracy in our libraries (i.e., the presence of isobaric compounds with different molecular structures), structure annotation based on MS/MS fragmentation spectra is essential for unequivocal compound identification (Fig. 3a). In the following, we will stick with the term "decoding" for annotating the query molecules, although there are no (sequence) tags involved here. To investigate the decodability of our libraries and simulate the final sample from an affinity selection, we prepared defined subsets of 245 to 500 compounds for each scaffold and analyzed each subset via nanoLC-MS/MS (the number of building blocks for miniSEL 1, 2, and 3 are shown in Fig. 3b and all structures are shown in Supplementary Fig. 14). Based on the known list of library compounds we analyzed the data and counted the detectable compounds. The number of detectable compounds is usually lower than the theoretical number due to numerous reasons, including loss during sample preparation of highly polar/unipolar structures, chromatography restrictions and compounds failing to ionize. Overall, each nanoLC-MS/MS run produced approximately 80,000 MS1 and MS2 scans, including mainly spectra resulting from background noise: for a real affinity selection sample of unknown content and complexity, the manual analysis of such a dataset is clearly impractical.

We first evaluated automated structure annotation using off-the-shelf metabolomics software. While typical metabolomics workflows use spectral databases as an input, our libraries representing novel chemical matter do not have such spectral databases. To this end, we used SIRIUS 6 and CSI:FingerID, considered best-in-class for reference spectra-free structure annotation of small molecules[37–39]. CSI:FingerID annotates compounds by scoring predicted molecular fingerprints against fingerprints of database structures (e.g., PubChem). In an affinity selection experiment, contrary to a regular metabolomics analysis, the complete space of potential structures is known, and the computationally enumerated library can be used as a structure database to score compounds against. To this end, we created custom structure databases in SIRIUS, consisting of the fully enumerated

library SMILES for each library (499,720, 216,008 and 31,800, respectively). We then imported the measured nanoLC-MS/MS runs into SIRIUS and performed a standard SIRIUS structure annotation workflow (Supplementary Chapter 4). While a good fraction of library compounds were correctly detected and annotated (82%, 77% and 81%, respectively for SEL 1, 2 and 3), the total number of annotated scans (up to 2800) largely exceeded the number of molecules actually present in the library (Fig. 3c). In principle, manual inspection can help picking out correct annotations but with thousands of false positives this off-the-shelf automated annotation workflow would still be impractical for real affinity selections samples with unknown content.

To increase the proportion of genuine library molecules in the final set of proposed compounds and to improve compound annotation, we established fragmentation rules to predict likely MS/MS patterns for our library compounds. We analyzed the fragmentation spectra of our test library compounds and calculated fragmentation frequencies of bonds connecting the various building blocks (Fig. 3d). For each of the three library scaffolds, we defined the most prominent recurring fragmentation modes. Based on these patterns, we generated a combinatorial "fragmenter" to create a list of predicted fragments for each library member (Fig. 3e). We implemented a filter, stipulating that only scans with an MS1 precursor mass matching a library compound's mass and containing at least one predicted fragment peak in their MS2 would be selected for full annotation (Fig. 3f). This filter drastically reduced the number of total scans (compare Fig. 3c, f), while maintaining a high correct recall and annotation rate of 66–74%. Overall, this Combinatorial Mass Encoding decoding Tool (COMET) enables the high-fidelity annotation and decoding of satisfying numbers of compounds from all three library scaffolds, with substantially reduced numbers of false positive annotations (a detailed description of COMET is reported in Supplementary Chapter 4).

## Selections against CAIX

With the three SELs in hand and the automated COMET software established, we initiated de novo ligand discovery experiments with the oncology drug target carbonic anhydrase IX (CAIX). CAIX has been used previously to benchmark novel DNA-encoded libraries due to its predictable binding profile to aromatic and heterocyclic sulfonamides[11,16,17]. CAIX is a therapeutically relevant target for cancer treatment, particularly in hypoxic tumors, due to its role in tumor cell survival and pH regulation in the tumor microenvironment[40]. We immobilized biotinylated CAIX on streptavidin-coated magnetic beads and incubated it with the half-million-membered SEL 1. After washing away non-binders, we eluted potential hits with $H_2O$/MeCN/FA (50:50:0.01) and analyzed the resulting sample with our nanoLC-MS/MS COMET workflow. To exclude unspecific binders, we also ran the library against streptavidin-coated magnetic beads and implemented background subtraction into COMET that removes spectra from CAIX runs if they result from features present in the control runs (using a 3-fold intensity ratio). From the resulting annotated spectra, we plotted the building block frequencies for each position and found a substantial enrichment of 4-sulfamoylbenzoic acid in the carboxylic acid position (Fig. 4a). This finding aligns well with known carbonic anhydrase binders, where the aromatic sulfonamide forms a strong

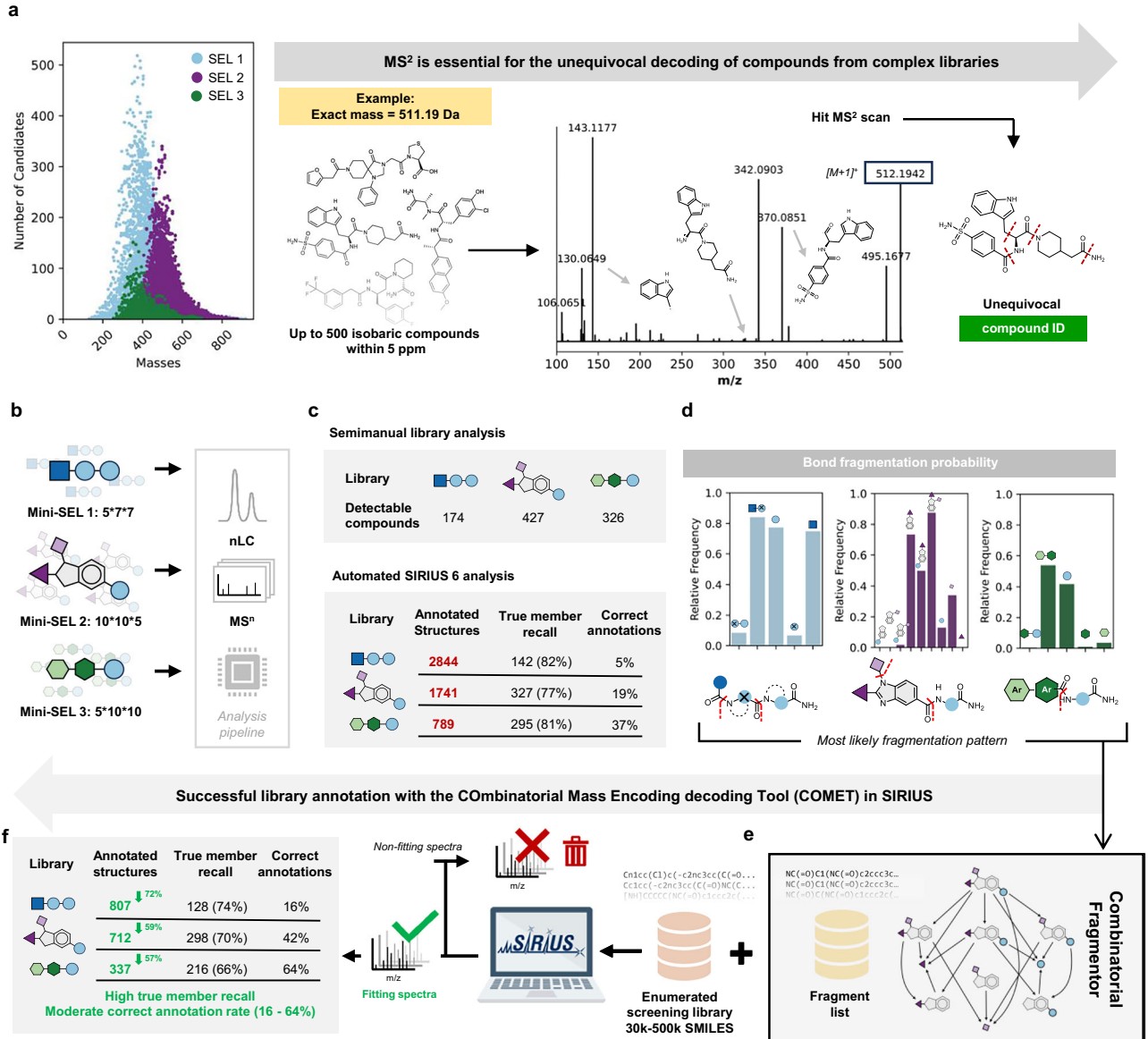

**Fig. 3 | Decoding with SIRIUS & COMET. a** The scatter plot shows the mass degeneracy of each library design. For specific masses there can be up to 500 possible different compounds. Tandem mass spectrometry is essential to enable the unequivocal annotation of unique structures. **b** For each library scaffold, a small combinatorial sublibrary with 245 or 500 different compounds was synthesized and measured by LC-MS/MS to mimic the outcome of an affinity selection outcome. **c** Given that in these libraries all compounds are known, we performed a semi-manual analysis to obtain the actual number of detectable compounds. Given that each nanoLC-MS/MS run produced ~ 80,000 spectra, such manual analysis would be unfeasible for samples with unknown content. Automated analysis with off-the-

shelf SIRIUS 6 resulted in good recalls of library compounds, but in very high rates of false positives (annotated compounds that are not actually present in the library). **d** Based on the manually annotated test libraries, fragmentation frequencies of each library scaffold were calculated and represented as histograms. **e, f** The COMET workflow consists of generating a fragment list based on the calculated fragmentation frequencies, filtering MS/MS scans based on the minimum number of matching peaks and structure annotation. The COMET workflow results in a substantial decrease in falsely annotated structures while retaining high recall rates of correct compounds. Source data are provided as a Source Data file.

interaction with a zinc ion in the binding pocket[41]. Among the 228 annotated structures, 74 different hits contained this substructure, and these compounds showed clear extracted ion peaks in the CAIX sample, while they were not detectable in the control runs (see Fig. 4b for selected examples). Fisher's exact test shows that this enrichment of a specific building block has a strong statistical significance, with a *p*-value of approximately $10^{-97}$ (Supplementary Fig. 18).

We also performed the selection with SEL 2 and 3 and identified aromatic sulfonamides as the most enriched BBs: 30 out of 75 and 47 out of 51 total hits, for SEL 2 and 3, respectively, contained that building block, both with statistical significance (Supplementary

Fig. 18). Adjacent positions also showed preferences for specific BBs (Fig. 4a), indicating the combination of the aryl sulfonamide with these BBs might result in preferred scaffolds for CAIX binding. A more stringent COMET filter matching at least two predicted fragments in the MS2 resulted in fewer annotated structures but decreased the number of hits for SEL 2 and SEL 3 (Supplementary Table 1). We did not observe any specific enrichment of BBs prone to stronger ionization, such as lysine or arginine, with their positive charges.

From the three different libraries, we selected hits for resynthesis and binding validation. All compounds demonstrated low nanomolar binding to CAIX, as tested by biolayer interferometry (Fig. 4b). Taken

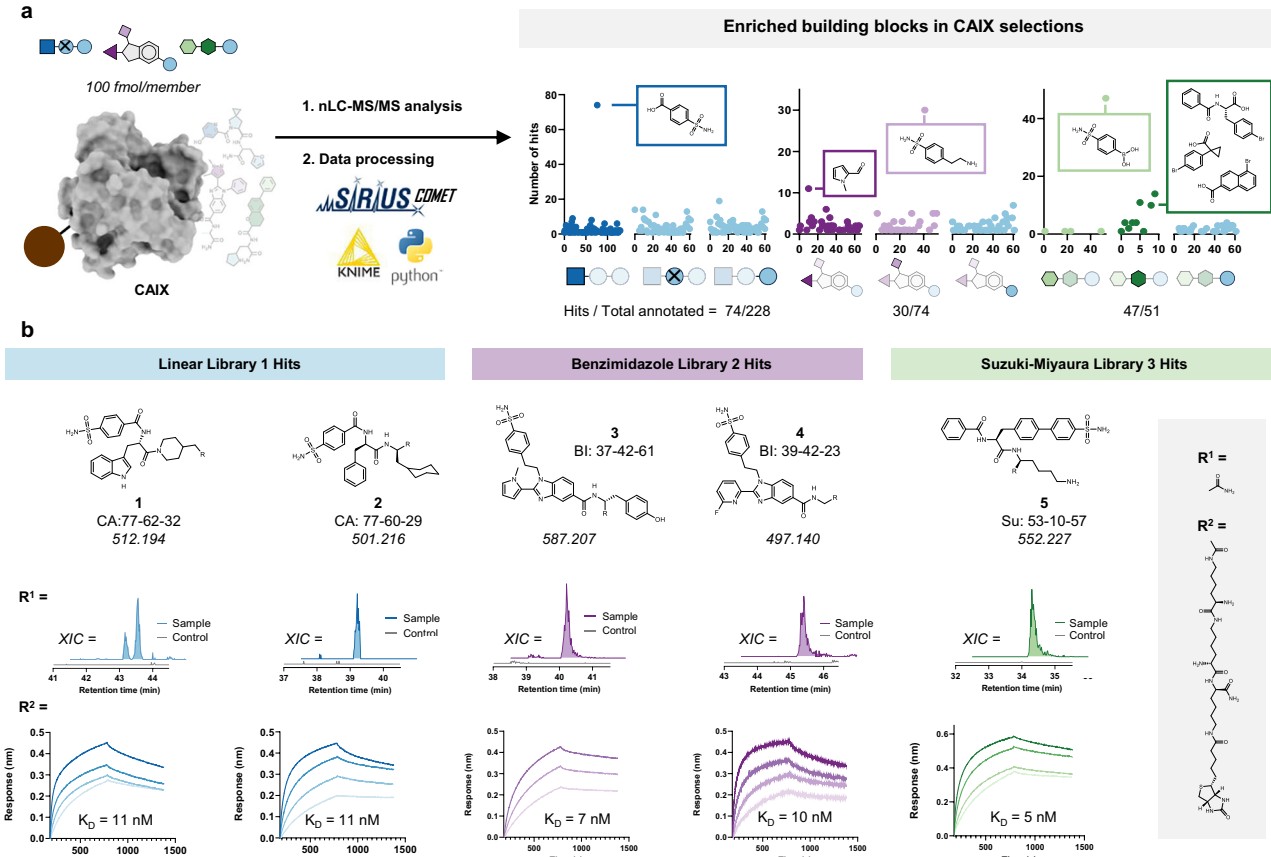

**Fig. 4 | Identification and validation of ligands against CAIX. a** SEL 1, 2 and 3 were panned in duplicates against immobilized CAIX. While MS1 features were consistently detected in both duplicates, annotated hits showed only a moderate overlap (Supplementary Figs. 16, 17). After analysis of the selected samples via nanoLCMS/MS and COMET, we plotted building block frequencies for each library scaffold and position. Aromatic sulfonamides were significantly enriched from all libraries ($p < 0.0001$, see Supplementary Fig. 18). **b** Examples of hit compounds: the chromatograms show extracted ion counts using a 5 ppm accuracy of their corresponding masses in the CAIX sample and bead-only control, demonstrating their selective enrichment. Biotinylated versions of these examples were resynthesized, and BLI association/dissociation curves confirmed their binding to CAIX. Source data are provided as a Source Data file.

together, our platform enables the selection of high-affinity binders from libraries with diverse scaffold architecture.

When screening diverse libraries by affinity selections, achieving high enrichments, i.e., removing non-binders from the final pool of selected compounds, and therefore increasing the binder/non-binder ratio compared to the full library, is essential. The enrichment can be calculated as follows:

$$\frac{\text{\# found binders}/\text{\# compounds after selection}}{\text{\# total binders}/\text{\# total library}} \quad (1)$$

Given that the actual total number of binders present in the library is unknown, usually the maximum enrichment is calculated, postulating the number of found binders equal to the total number of binders[30,31]. The enrichments obtained in our three library selections are $2.2 \times 10^3$ ($\frac{74/228}{74/499720}$), $2.9 \times 10^3$ ($\frac{30/75}{30/216008}$) and $6 \times 10^2$ ($\frac{47/51}{47/31800}$), respectively, for SEL 1, 2 and 3. It is noteworthy that SEL 3 achieved an almost perfect enrichment, as 47 of the 51 identified scans contain binders. These enrichment scores are in good agreement with typical phage display and peptide AS-MS selections, which also achieve enrichments in the order of magnitude of $10^{32,42}$.

We next investigated the effect of library concentration on ligand identification. Considering the solubility limits of any library, a lower initial concentration of each member for the affinity selection would enable the screening of larger libraries. We performed the selection of CAIX ligands with SEL1 at 1 pmol/member, 100 fmol/member and 10

fmol/member. For 1 pmol/member to 100 fmol/member, there is only a 1.3-fold decrease in the absolute number of hits. Lowering the concentration to 10 fmol/member, no hits were recovered (Supplementary Table 2). Given that the selection results in similar outcomes at 1 pmol/member and 100 fmol/member, it indicates that we might be able to run affinity selections with 10 times larger libraries, i.e., 5 million members, using 100 fmol/member. Even larger library sizes are likely to cause solubility limitations.

To explore whether the hit discovery process could be further accelerated and streamlined, we conducted affinity selection using a pooled combination of all three libraries. This combined library encompasses approximately 750,000 members, representing a higher degree of molecular diversity. Encouragingly, sulfonamide-based CAIX binders were identified across all three library scaffolds, with 90 hits found in total. Although the pooled approach yielded fewer hits compared to individual library selections (74 + 30 + 47 = 151 hits), our experiment demonstrates the potential for applying this workflow to high-diversity libraries with varied scaffold architectures.

## Selections against FEN1

After establishing a general workflow for SEL selections and validating its potential with CAIX, we sought to explore its applicability to targets beyond the scope of DEL screenings. To this end, we applied our workflow to identify inhibitors for flap endonuclease 1 (FEN1), a DNA-processing enzyme essential for replication and repair. FEN1 is over-expressed in multiple cancer types, and its synthetic lethality

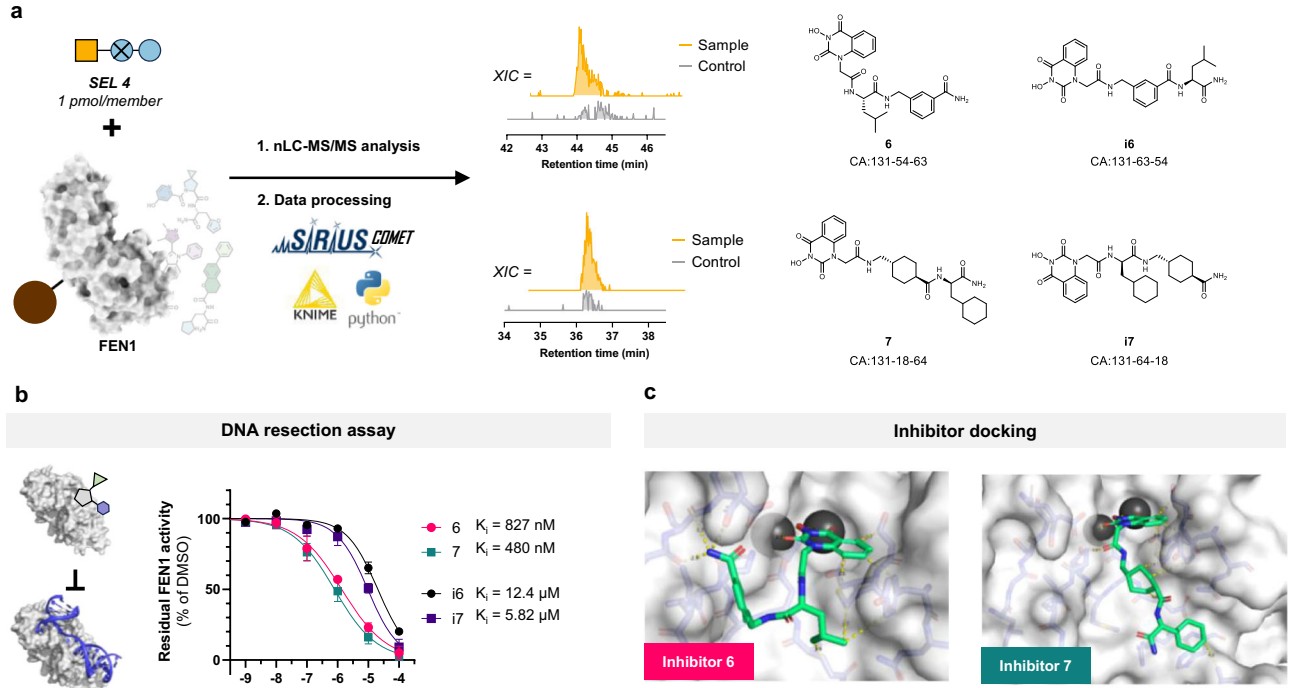

**Fig. 5 | Identification and validation of ligands against FEN1. a** SEL 4 was screened against immobilized FEN1. In the affinity selection against FEN1, two clear hits were identified. The chromatograms show extracted ion counts using a 5 ppm accuracy of their corresponding masses in the FEN1 sample and bead-only control, demonstrating their selective enrichment. MS/MS fragmentation could be assigned to four compounds that were selected for resynthesis. **b** Resynthesized compounds were tested for inhibition against FEN1 in a fluorescence polarization based DNA resection assay. **c** Docking studies on **6** and **7** show the coordination of the *N*-hydroxyurea BB to the magnesium ions and hydrophobic interactions of the Leucine/Cyclohexyl building blocks with Met37 and Tyr40. Source data are provided as a Source Data file.

interactions with frequently mutated cancer genes make it a promising therapeutic target[43].

Our initial selections using SELs 1–3 did not yield clearly enriched BB patterns, nor did we detect individual MS features that could be confidently assigned to enriched library compounds. The relatively shallow and open binding site of FEN1, which evolved to accommodate double-stranded DNA, may pose a challenge for identifying high-affinity small molecules. To enhance the selection process, we designed SEL 4, a focused 4000-member library based on the architecture of SEL1 but enriched with BBs designed to favor binding to the FEN1 active site. Specifically, we incorporated nucleobase analogs to mimic the DNA substrate and functional groups from known nuclease inhibitors, such as *N*-hydroxyureas and trihydroxy phenols[44,45].

The selection led to the identification of two compounds, **6** and **7**, which were substantially enriched over background (3.8 and 3.9 fold, respectively, Fig. 5a). Tandem MS spectra confirmed the presence of an *N*-hydroxyurea BB at position 3, a previously described functional group known to coordinate the two magnesium ions in the FEN1 binding pocket[45]. Since the exact positioning of the additional two building blocks could not be assigned with absolute confidence, we also synthesized their inverted variants, **i6** and **i7**.

All four compounds were tested in a FEN1 DNA resection assay. **6** and **7** inhibited FEN1 activity with $IC_{50}$ values of 827 nM and 480 nM, respectively, whereas **i6** and **i7** displayed 10- to 15-fold weaker activity (Fig. 5b). Docking studies with **6, 7**, having the *N*-hydroxyurea BB coordinating the two-magnesium-ion core, suggested that the adjacent aliphatic BBs likely form hydrophobic interactions with Met37 and Tyr40 within the binding groove (Fig. 5c).

Taken together, the SEL-COMET workflow succeeded in aiding the selection and identification of FEN1 inhibitors, directly targeting its DNA-binding site.

## Discussion

Herein, we have described the use of barcode-free self-encoded libraries (SELs) for affinity selection-based hit discovery. Our approach leverages tandem MS fragmentation to accurately reconstruct the molecular structure of compounds selected from vast libraries, eliminating the need for large barcodes as required by previous platforms like DELs. The SEL strategy achieves the maximal possible information density for library selections (defining information density as [molecule mass]/[molecule + decoding tag mass]), significantly larger than commonly used DNA-encoded molecules. With this feature, each molecule is present in the library in its completely unmodified form, abrogating any bias resulting from a large encoding tag. To support the broad validity of our strategy, we have tested the decoding on three different library architectures: we succeeded in efficiently decoding compounds from all three scaffolds via tandem MS in an automated fashion. The diversity in chemical connectivities and the drug-likeness of our libraries are of great promise for the potential expansion of our technology to many more interesting library architectures. We also performed affinity selection with libraries containing up to 750 thousand members against CAIX and with 4 thousand members against FEN1. In these selections, we decoded and validated several nanomolar binders for CAIX and inhibitors for FEN1, demonstrating the applicability of our methodology to drug discovery campaigns.

SELs offer considerable advantages in synthesis complexity and scope compared to DELs, which require an alternation of enzymatic ligation steps and chemical reactions performed on the oligonucleotide-linked scaffold. While an impressive number of compatible reactions have been developed[18], there are always concerns of potential degradation of the DNA barcode[19]. Roessler et al. recently reported the DNA lability under a number of conditions, and showed,

on the other side, how peptides synthesized on a solid support can withstand many harsh conditions, required for small molecule synthesis[19]. In that study, the authors point out the advantages of peptides as encoding tags of small molecules. The synthesis of peptide-encoded libraries (PELs) requires more than 40 reaction steps, in addition to the utilization of orthogonal cleavable linkers. SELs can be synthesized in as few as five straightforward reaction steps on solid supports. Because of the absence of an encoding tag, SELs can undergo any reaction condition compatible with the small molecule itself. We successfully demonstrated several transformations, including amide bond formations, nucleophilic aromatic substitutions, heterocyclizations, palladium-catalyzed cross couplings, and acid and base-mediated protecting group removals. We conjecture that many more transformations may be applied to prepare SELs in the future.

Notably, high-diversity SELs derived from our library scaffolds can be synthesized in under a week using standard organic synthesis techniques, making this approach accessible to almost any laboratory. Given that many institutions have mass spectrometry facilities, our streamlined synthesis, selection, and analysis workflow positions SELs to democratize rapid early-stage drug discovery.

SELs remain subject to a key limitation inherent to combinatorial libraries, namely, low scaffold diversity within individual libraries. However, this can be mitigated by the ability to rapidly generate distinct combinatorial libraries, thereby expanding the accessible chemical space. While complex polycyclic scaffolds may exceed the current capabilities of SEL decoding, many pharmaceutically relevant scaffolds could, in principle, be decoded using SIRIUS-COMET. Although generating very diverse libraries is more demanding than those based on a single scaffold, existing HTS libraries could be potentially pooled and enable rapid affinity selections, significantly streamlining the early phases of hit discovery.

The absence of a barcoding tag in SELs can offer substantial advantages during affinity selection. Barcoding tags, often larger than the molecules themselves, can interact undesirably with the target, leading to false positives, especially in DEL selections involving nucleic acid-binding proteins. Moreover, the bulky tags in DELs and PELs limit the conformational flexibility of library molecules, restricting how they can bind to their targets. SEL compounds, free from these constraints, avoid these pitfalls and potentially sample a much broader conformational diversity during affinity selection. In addition, misencoding can be circumvented: with the synthesis of barcoded compounds proceeding with distinct steps for barcode and molecule construction, any deletion or truncation on either part of the construct will not correspond to the corresponding entity on the other part of the molecule, leading to a mismatch between actual molecule and barcode[17,46]. Self-encoded molecules, self-containing all the information about their structure, obviate such problems.

With SEL library sizes in the range of one million compounds, we match the scale of several recent successful DELs[11,12]. While DEL libraries can theoretically reach billions of compounds, recent evidence suggests that DEL selections work best with inputs exceeding $10^6$ individual copies per compound, effectively capping optimal library sizes at a few million members[16,47]. Unlike DELs, SEL hits cannot be amplified, so the input quantity for a selection must account for the sensitivity limits of the mass spectrometer used for detection and decoding. However, modern instruments can typically detect as little as ~1 fmol per molecule, or even less. For our experimental setup, we found that 100 fmol per molecule is an ideal input for affinity selection, in principle allowing for screenings with at least 5 million compounds simultaneously, remaining within the library solubility limits.

In large libraries, mass degeneracy is inevitable, necessitating the development of computational methods for automated decoding of library members and affinity selection hits. Our COMET

workflow was developed to address this challenge. We do not rely on spectral databases for high-confidence compound identification, since synthetic SELs lack such databases; this necessitates in silico compound annotation. The COMET workflow enables the correct annotation of library compounds with acceptable recall rates and low numbers of false positive annotations. The automated computational analysis of the experimental data is instrumental for investigating complex affinity selection samples resulting from the screening of large libraries. Notably, we observed an increase in false positive annotations with larger library sizes. COMET effectively reduced the number of proposed structures to a manageable level, but at this last stage, manual analysis of the proposed candidates and their MS/MS spectra was necessary. Further improvements in compound filtering and candidate ranking will be essential to enable large-scale screenings.

In affinity selections against CAIX, we validated the potential of our SEL COMET platform for rapid hit discovery, identifying multiple nanomolar binders across all library scaffolds. The CAIX protein, with its affinity for aromatic sulfonamides, has previously proven as a useful model target for the validation of DELs[11,17]. The presence of multiple binders with defined molecular features within our SELs enabled us to gain valuable insights about achievable enrichment, optimal library input quantities, and the possibility of screening multiple libraries with diverse scaffolds simultaneously.

Finally, our study demonstrates that the SEL selection workflow can be successfully applied to DNA-processing enzymes, expanding its utility beyond traditional DEL screening targets. Despite the inherent challenges posed by the shallow and open binding site of FEN1, we were able to identify reasonably potent inhibitors, validating the approach for this class of targets. The rapid synthesis of SEL 4–a focused library tailored to FEN1–highlights the flexibility and efficiency of the SEL platform. This streamlined synthesis capability enables the rapid design and testing of target-specific SELs, facilitating the discovery of inhibitors for challenging enzymatic targets with minimal synthetic effort.

In summary, our findings demonstrate a barcode-free technology with large self-encoded libraries for early drug discovery. We anticipate that this approach will see widespread adoption in both academic and industrial research settings.

## Methods
Detailed methods, synthetic procedures, compound characterization, building block selection, library enumeration and COMET are described in the Supplementary Information.

### Library synthesis
**SEL 1.** TentaGel S NH$_2$ resin (30 µm, 0.24 mmol/g loading, 2.625 g, 630 µmol, 1.0 eq.) functionalized with Fmoc-Rink Amide linker was over-divided over 62 fritted syringes. A solution of each Fmoc-protected amino acid **AA1-AA62** (30 µmol, 3.0 eq), HATU (29.8 µmol, 0.4 M, 2.98 eq) and DIPEA (90 µmol, 9.0 eq) in DMF was added to the resin and reacted for 2 h. The resin was pooled and washed with DMF (5 × 2 mL) and 20% piperidine in DMF (1 × 2 mL) before incubating with 20% piperidine in DMF for 10 min. The resin was washed with DMF (5 × 2 mL) and split over 62 fritted syringes.

The second building block was incorporated using the same reaction conditions. A solution of each Fmoc-protected amino acid **AA1-AA62** (30 µmol, 3.0 eq), HATU (29.8 eq. 0.4 M, 2.98 eq) and DIPEA (90 µmol, 9.0 eq) in DMF was added to the resin and reacted for 2 h. The resin was pooled into a fritted syringe (20 mL) and washed with DMF (5 × 2 mL) and 20% piperidine in DMF (1 x 2 mL) before incubating with 20% piperidine in DMF for 10 min

For the incorporation of the 3$^{rd}$ building block, the resin (350 µmol, 1.0 eq) was divided over 130 Eppendorf tubes. A solution of each carboxylic acid **CA1-CA130** (8.08 µmol, 3.0 eq), HATU (80.2 µL,

0.1 M, 8.02 µmol, 2.98 eq) and DIPEA (4.22 µL, 24.23 µmol, 9.0 eq) in DMF was added to the resin (2.69 µmol). The reactions were stirred overnight at r,t. The resin was pooled and washed with DMF (5 × 2 mL) and DCM (5 × 2 mL).

The resin was incubated for 1.5 h with a solution of TFA:H$_2$O:TIPS (92.5:5:2.5) and washed once with a solution of TFA:H$_2$O:TIPS (92.5:5:2.5). TFA was evaporated under a stream of N$_2$, and the library was purified using reverse phase column chromatography with a stepwise gradient of 00-70-100% MeCN:H$_2$O (0.1% TFA).

**SEL 2.** TentaGel S NH$_2$ resin (30 µm, 0.24 mmol/g loading, 2.625 g, 630 µmol, 1.0 eq.) functionalized with Fmoc-Rink Amide linker was over-divided over 62 fritted syringes. A solution of each Fmoc-protected amino acid **AA1-AA62** (30 µmol, 3.0 eq), HATU (29.8 µmol, 0.4 M, 2.98 eq) and DIPEA (90 µmol, 9.0 eq) in DMF was added to the resin and reacted for 2 h. The resin was pooled and washed with DMF (5 × 2 mL) and 20% piperidine in DMF (1x, 2 mL) before incubating with 20% piperidine in DMF for 10 min. The resin was washed with DMF (5 × 2 mL) and a solution of 4-fluoro-3-nitrobenzoic acid (175 mg, 945 µmol, 3.0 eq), HATU (2.346 mL, 939 µmol, 2.98 eq) and DIPEA (494 µL, 2.835 mmol, 9.0 eq) in DMF was added to the resin. After 1 h, the reaction was washed with DMF (5 × 2 mL). The resin was divided over 52 Eppendorf tubes, to which amines **AM1-AM52** (59.6 µmol, 10 eq.) and DIPEA (10.55 µL, 59.6 µmol, 10 eq.) in DMF (150 µL, 0.4 M) were added. The mixture was shaken at 1 x *g* overnight at 80 °C. The resin was pooled and washed with DMF (5 × 2 mL) and DCM (5 × 2 mL). A solution of 1.0 M SnCl$_2$ (3.73 g, 19.7 mmol, 62.5 eq.) in DMF was added to the resin (315 µmol). The mixture was incubated overnight at r.t., whereafter the resin was washed with DMF:H$_2$O (1:1, 5 × 2 mL), with DMF (5x, 2 mL) and DCM (5x, 2 mL). The resin (4.31 µmol, 1.0 eq.) was split over 67 Eppendorf tubes to which the appropriate aldehyde **AL1-AL67** (0.25 M in DMF (103.3 µL), 25 µmol, 5 eq.) and p-TsOH•H$_2$O (4.76 mg, 25 µmol, 5 eq.) were added. The mixture was incubated overnight at 1 x *g*, overnight at r.t. After incubation, the resin was washed with DMF (5 × 2 mL) and DCM (5 × 2 mL). The resin was incubated for 1.5 h with a solution of TFA:H$_2$O:TIPS (92.5:5:2.5) and washed once with a solution of TFA:H2O:TIPS (92.5:5:2.5). TFA was evaporated under a stream of N$_2$, and the library was purified using reverse phase column chromatography with a stepwise gradient of 00-70-100% MeCN:H$_2$O (0.1% TFA).

**SEL 3.** TentaGel S NH$_2$ resin (30 µm, 0.24 mmol/g loading, 1.1 g, 265 µmol, 1.0 eq.) functionalized with Fmoc-Rink Amide linker was over-divided over 60 fritted syringes. A solution of each Fmoc-protected amino acid (Supplementary Table 12) (12.8 µmol, 3.0 eq), HATU (12.7 µmol, 0.4 M, 2.98 eq) and DIPEA (38.5 µmol, 9.0 eq) in DMF was added to the resin and reacted for 4 h. The resin was pooled into a fritted syringe (20 mL) and washed with DMF (5 × 2 mL) and 20% piperidine in DMF (1x, 2 mL) before incubating with 20% piperidine in DMF for 10 min. The resin was divided over 10 Eppendorf tubes. A solution of aryl bromide **AB1-AB10** (26.50 µmol, 3.0 eq), 0.4 M HATU (78.97 µmol, 197 µL, 2.98 eq), DIPEA (41.50 µL, 238.5 µmol, 9.0 eq) and DMF (400 µL) was added to the resin (26.50 µmol). After 4 h, the resin was washed with DMF (5 × 2 mL). Aryl bromide **AB10** was deprotected by washing with 20% piperidine in DMF (1x, 2 mL) before incubating with 20% piperidine in DMF for 10 min. A solution of benzoic acid (9.7 mg, 79.50 µmol, 3 eq.), DIPEA (41.50 µL, 238 µmol, 9.0 eq) and DMF (400 µL) was added to the resin containing **AB10** (26.50 µmol) and reacted for 1 h. The resin was combined in a fritted syringe and was washed with DMF (5x). The resin was divided over 53 Eppendorf tubes. Boronic acid **BA1-BA53** (10 µmol, 2.0 eq.), K$_2$CO$_3$ (1.4 mg, 10 µmol, 2.0 eq.), PdCl$_2$ (89 µg, 0.5 µmol, 10 mol%), XPhos (0.48 mg, 1 µmol, 20 mol%) and DMF (100 µL) were added to the resin (5.0 µmol). The reaction was stirred at 1 x *g* at 80 °C for 21 h. The resin was washed with DMF (5 × 2 mL)

and DCM (5x) and incubated for 1:45 h with a solution of TFA:H$_2$O:TIPS (92.5:5:2.5). The resin was washed once with a solution of TFA:H$_2$O:TIPS (92.5:5:2.5), whereafter the TFA was evaporated. The library was purified using reverse phase column chromatography with a stepwise gradient 00-70-100% MeCN (0.1% TFA).

**SEL 4.** TentaGel S NH$_2$ resin (90 µm, 385 mg, 0.26 mmol/g, 100 µmol, 1.0 eq.) functionalized with Fmoc-Rink Amide linker was over-divided over 20 fritted syringes. A solution of each Fmoc-protected amino acid (Supplementary Table 15) (15 µmol, 3.0 eq), HATU (0.4 M, 2.98 eq) and DIPEA (9.0 eq) in DMF was added to the resin and reacted for 2.5 h. The resin was pooled and washed with DMF (5 × 2 mL) and 20% piperidine in DMF (1 × 2 mL) before incubating with 20% piperidine in DMF for 10 min. The resin was washed with DMF (5 × 2 mL) and split over 20 fritted syringes. The second building block was incorporated using the same reaction conditions. A solution of each Fmoc-protected amino acid (Supplementary Table 15) (15 µmol, 3.0 eq), HATU (0.4 M, 2.98 eq) and DIPEA (9.0 eq) in DMF was added to the resin and reacted for 2.5 h. The resin was pooled into a fritted syringe (20 mL) and washed with DMF (5 × 2 mL) and 20% piperidine in DMF (1x, 2 mL) before incubating with 20% piperidine in DMF for 10 min

For the incorporation of the 3$^{rd}$ building block, the resin was divided over 10 syringes. A solution of each carboxylic acid (Supplementary Table 16) (30 µmol, 3.0 eq), HATU (149 µL, 0.2 M, 29.80 µmol, 2.98 eq) and DIPEA (15.7 µL, 90 µmol, 9.0 eq) in DMF was added to the resin (10 µmol) and reacted for 2.5 h. The resin was pooled and washed with DMF (5 × 2 mL) and DCM (5 × 2 mL).

The resin was incubated for 1.5 h with a solution of TFA:H$_2$O:TIPS (92.5:5:2.5) and washed once with a solution of TFA:H$_2$O:TIPS (92.5:5:2.5). TFA was evaporated under a stream of N$_2$, and the library was purified using reverse phase column chromatography with a stepwise gradient of 00-70-100% MeCN:H$_2$O (0.1% TFA).

**Affinity selection mass spectrometry.** A KingFisher™ Duo Prime Purification System was used to perform our affinity selection experiments. The protocols were developed with BindIt 4.1 Software. Affinity selection experiments against CAIX were performed in duplicates with protein (150 pmol) immobilized on Dynabeads MyOne Streptavidin T1 (1 mg) and library (100 fmol/member) with the King Fisher protocol as described in Supplementary Fig. 19, unless stated otherwise. Affinity selection experiments against FEN1 were performed in duplicates with protein (150 pmol) immobilized on Dynabeads™ His-Tag Isolation and Pulldown (1 mg) and library (1 pmol/member) with the King Fisher protocol as described in Supplementary Fig. 19.

Samples from the affinity selection procedure were lyophilized and resuspended in 50 µL MQ 0.1%FA. The StageTips were prepared as described by Rappsilber et al. using C18 material from Empore SPE 47 mm disks (66883-U, Merck)[48]. The StageTips were pre-conditioned with 200 µL MeOH, 200 µL of 0.1% (v/v) FA in MeCN and 200 µL of 0.1% (v/v) FA in MQ, respectively, by centrifuging for 3 min at 300 x *g*. The samples were then loaded on the StageTips and washed with 200 µL of 0.1% (v/v) FA in MQ. Compounds were eluted by adding 200 µL of 0.1% (v/v) FA in MeCN:MQ (7:3). The samples were lyophilized before resuspending in 10 µL 0.1% (v/v) FA in UPLC-MS grade water. The samples were centrifuged for 5 min at 21,000 x *g*. Afterwards, 9 µL was transferred to a LC-MS vial and 8 µL was injected into the LC-MS/MS system. Additional details are reported in Supplementary Chapters 8 and 9.

**COMET.** COMET software is available through GitHub (https://github.com/sirius-ms/comet). Enumerated libraries generated by the above KNIME workflow were imported into COMET as custom structure databases using the GUI. Spectra files were imported as.mzML files and

background subtraction was performed using the "Tags" panel in the general filter dialog window of the GUI using the following settings; MS1 m/z accuracy = 5 ppm, RT accuracy = 10 s and max intensity ratio = 3 (see Supplementary Fig. 20). Features from actual samples that also appeared in the control runs within a two second retention time tolerance, a five ppm mass deviation tolerance, and a fold change of less than two were removed. The COMET filter can be accessed through the same filter dialog window, where a separate section for COMET is provided (Supplementary Fig. 20). For its application, we used the following settings: scaffold formula: C8H3N2O for SEL 2 and left blank for SEL 1, 3 and 4, MS1 mass accuracy (ppm) = 5 ppm, considered fragment types = SEL 1: S[0;1], S[1;2],0,2, SEL 2: S[0;2], S[1;2],0,1, SEL 3: S[1;2],0, SEL 4: S[0;1], S[1;2],0,2 minimum number of matching peaks = 1, number of considered peaks = 5, number of allowed hydrogen shifts = 1, MS2 mass accuracy (ppm) = 5. The specification of such fragment types is illustrated in Supplementary Fig. 22. For each library, a file containing information about all the library's building blocks has to be provided in COMET. These.csv files were generated using the Python script "COMET_Building blocks_input.ipynb" which is available through Zenodo (https://doi.org/10.5281/zenodo.14070388). See Supplementary Chapter 4.1 for method details on the COMET filters. Molecular formula generation in COMET was performed using formula database search in the imported custom library; all other settings were left to default. After fingerprint prediction (score threshold enabled), the imported custom library was used for structure database search. Structure candidates were ranked according to EPIMETHEUS, see Supplementary Chapter 4.5, for method details.

**Biolayer interferometry (BLI).** Purified biotinylated compounds were dissolved to 1 µM in 1x PBS, 0.02% Tween-20, 1 mg/ml BSA (0.1% (w/v)) (kinetic buffer) used for immobilization onto streptavidin Octet SA Biosensors (SATORIUS). Biolayer interferometry (BLI) assays were performed in 96-well plates (GreinerBio-One, polypropylene, flat-bottom, chimney well) using an Octet R4 system (SATORIUS). Wells were filled with 200 µL of kinetic buffer, compound solution or CAIX solution.

Biotinylated compound was immobilized onto the streptavidin biosensor for 60 s. Sensors were then dipped into kinetic buffer for 60 s, CAIX solution (500 nM, 250 nM, 125 nM, 62.5 nM) for 600 s and into kinetic buffer for 600 s. Measurements were carried out at 30 °C.

**FEN1 activity assay.** The assay was performed based on literature[49]. The assay was performed in Corning black 384-well plates (no. 3820). The assay buffer contained 20 mM HEPES pH 7.5, 100 mM KCl, 5 mM $MgCl_2$, 0.1 mM DTT, 200 µg/mL BSA and 0.01% NP-40. The substrate consists of an oligonucleotide with Cy3 as a fluorophore. In each well, substrate (200 nM), FEN1 (125 pM) and were incubated with decreasing concentration of hits. The FP signal was read on a BMG PHERAstar.

**Docking.** Using published ligand-bound x-ray crystal structure of FEN1 (PDB: 5FV7) in pharmacophore modeling tool 'Pharmit'[50], four pharmacophore interactions were defined in the bound ligand. These interactions served to constrain ligands to form comparable interactions with the 2 x $Mg^{2+}$ ions, while no constraint would be placed on the orientation or position of the additional building blocks present in the hit. Two H-bond acceptors were defined for the carbonyls of the hydroxyurea. The hydroxy group was assigned as an H-bond donor (modeling was performed with the hydroxyurea as a neutral species). Finally, the urea-containing 6-membered ring was defined as an aromatic pharmacophore. Next, conformations for ASMS hits **6** and **7** were generated in Pharmit and conformers were aligned with the prepared pharmacophore model. >40 conformations were found as matches. Next, an energy minimization step was performed, and an

energy score filter (<−6) and maximum mRMSD (4.0) was set to eliminate low-quality poses.

**Reporting summary**
Further information on research design is available in the Nature Portfolio Reporting Summary linked to this article.

## Data availability
Raw and processed data used for software development and raw LC-MS/MS files have been deposited in the Zenodo database (https://doi.org/10.5281/zenodo.14070388). Source data are provided with this paper. All other data is available in the main text, the supplementary materials and from the corresponding author(s) upon request. Source data are provided with this paper.

## Code availability
COMET software is available through GitHub (https://github.com/sirius-ms/comet). COMET version 1.0.0 used in this study has been deposited in the Zenodo database under accession code https://doi.org/10.5281/zenodo.17225666.

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

## Acknowledgments

NAH, SB, MAH and MEL are supported by the Thüringer Ministerium für Wirtschaft, Wissenschaft und Digitale Gesellschaft (TMWWDG) with funds from the European Union as part of the European Regional Development Fund (ERDF, 2023 VFE 0003 and 0029). SJP acknowledges funding via the ERC Starting Grant (101039354). The Pomplun Lab gratefully acknowledges financial support from Mr. H. J. M. Roels through a donation to the Oncode Institute and KWF's financial support of the Oncode Institute.

## Author contributions

Conceptualization: S.J.P., S.B., Evd.N., and N.A.H. Methodology: S.J.P., S.B., Evd.N., N.A.H., M.A.H., M.E.L., M.L., J.M.M., O.B., and Gv.W. Synthesis: Evd.N., Q.Q.G., B.S., and S.Mc.K. Assays: Evd.N., S.M.N., and T.J.W. Software: N.A.H., M.A.H., M.E.L., M.L., and S.B., Visualization: Evd.N., N.A.H., J.M.M., and S.J.P. Funding acquisition: M.A.H., M.L., S.B., and S.J.P. Supervision: S.B. and S.P. Writing – original draft: Evd.N. and S.J.P. Writing – review & editing: Evd.N., N.A.H., S.J.P., and S.B.

## Competing interests

SJP, SB, EvdN, NAH and MAH have filed a patent application for the methodology described here. The remaining authors declare no competing interests.
