## [Transparent Peer Review file · Nature Communications]

Barcode-free hit discovery from massive libraries enabled by automated small molecule structure annotation

Corresponding Author: Dr Sebastian Pomplun

Version 0:

Reviewer comments:

Reviewer #1

(Remarks to the Author)

I am really impressed with the manuscript by van der Nol et al. For the computational aspects (EPIMETHEUS), I did not find them very innovative, as very similar in silico fragmentation approaches have been previously proposed. What really impressed me was the innovative application of these computational methods to high-throughput screening, with very detailed and unbiased explanation of bottlenecks (e.g. false positives) and strength of the approach. Additionally, the manuscript is very well written, and I believe the strengths of this manuscript and its innovativeness in application of in silico database search methods overcome its pitfalls. Please see my detailed comments below:

Major comments:

- Has 4-sulfamoylbenzoic acid (individually) or 4-sulfamoylbenzoic acid containing molecule been previously reported to interact with the target? Why is there a need to test half a million molecules, if similar results could be achieved by just screening a small number of building blocks including 4-sulfamoylbenzoic acid?

>> Supplementary material by breaking either a single nonring bond or a pair of bonds within the same ring system.

This is exactly how Dereplicator+ fragmentation graphs are constructed (single nonring bonds are called bridges and pair of bonds within the same ring system are called 2-cuts). Some of the other existing fragmentation methods might also be using a similar concept.

>> We filtered a comprehensive building block (BB) catalog (Fmoc-amino acids and carboxylic acids from Chemspace) by availability and price, narrowing it down to 1000 BBs per position.

What was the number before narrowing down

>> With these we enumerated a virtual library with a billion members. Is it $1000 \times 1000 \times 1000$?

Please explain where "a billion" came from ?

>> we generated a half million-membered

Is half a million referring to 499720? If so, use the latter for clarity

- Add a table and explain numbers 130, 62, 52, 67, 53, 10 and 60. Where each one is coming from. Its nice to have them all together.

>> 500k, 200k and 30k, respectively

Use exact numbers for clarity

>> we prepared defined subsets of 245 to 500 compounds for each scaffold and analyzed each subset via nanoLC-MS/MS

Please clarify exactly how many from each set

>> We implemented a filter, stipulating that only scans with an MS1 precursor mass matching a library compound's mass and containing at least one predicted fragment peak in their MS2 would be selected for full annotation (Fig. 3f).

How specificity / sensitivity changes if 2, 3, etc. predicted fragments are required, instead of only 1 ?

>> Discovery of ligands from high diversity SELs.
Selections against CAIX.

Seems like one subsection was left empty

>> 74 different hits contained this substructure

How many hits did you get in total ? How many would you see by random ? What is the statistical significance ? (e.g. using fisher p-value)

>> 30 and 47 individual compounds, for SEL 2 and 3, respectively

Out of how many ? How many would you see by random ? What is the statistical significance ?

>> 2.2×10^3 (74/228:74/499,720), 2.9×10^3 (30/75:30/216,008) and 6×10^2 (47/51:47/31,800)

I am a little confused, so #found binders = #total binders in all cases ? What does it mean? can you explain ?

>> 74/228:74/499,720

What is 228 ?

>> $[(\# \text{ found binders})/(\# \text{ compounds after selection})]:[(\# \text{ total binders})/(\# \text{ total library})]$.

Can you improve the presentation of this formula, aesthetically ?

>> 90 vs. 150

150 is not consistent with any of the numbers you previously reported:

2.2×10^3 (74/228:74/499,720), 2.9×10^3 (30/75:30/216,008) and 6×10^2 (47/51:47/31,800)

>> we have tested the decoding on three different library architectures:

Three? or four ?

- Even after application of EPIMETHEUS threshold, false discovery is still high. I acknowledge going from half a million to only a few hundred FPs is still an impressive job. However, The authors should expand (at least in discussion) how these false positives will impact downstream analysis.

- It is well known in drug discovery that these sort of combinatorial libraries (Sel 1 – Sel 4) are less diverse and less interesting than more diverse alternatives. I was expecting a discussion of this, along with potential bottlenecks of applying the proposed method to more diverse libraries (it is just the issue of cost of library?)

(Remarks on code availability)

Reviewer #2

(Remarks to the Author)

The author present a new approach allowing the screening of large collections of small molecules by affinity selection without specific library encoding. Hit compounds are characterized by a MS1/MS2 fragment mass encoding. The strategy has been used and described for large peptide libraries but not for small molecules. The article is very well written and pleasant to read. Despite progresses described in the annotation, the percentage of compounds correctly annotated remains quite low to my perception and should be presented as such. The article is well documented and detailed information given to facilitate reproduction of the work.

Comments to improve the quality of the article:

-Abstract: "screening barcode-free libraries of this scale is unprecedented" Not really true. Larger libraries of peptides have been screened in AS-MS and multi-million small molecule collections screened using pooling strategies.

-Page 7, ligne 41: How do you check that compounds are annotated correctly?

-Figure 3: "sublibrary with less than 500 unique compounds" Unique compounds refers usually to single compounds. I understand that those compounds are produced as mix of compounds.

-Fig3b : The table presented to illustrate the COMET annotation should be comparable to the Automated SIRIUS 6 analysis table. Data should be presented in the same order, that is, Annotated structures, true member recall and Correct annotations (missing column on purpose?). The correct annotation rate is only 16% for library SEL1 and 41 % for SEL2, 64% for SEL 3. Despite the improvement obtained as compared to the automated SIRIUS 6 analysis, the values obtained for SEL 1 & 2 are far from being a "high correct annotation rate". I will recommend to use "Moderate correct annotation rate"

-Page 11, ligne 24: "which were substantially enriched over background" Could you be more specific and state the enrichment cut-off ratio used to separate hits from background.

-Page 13, ligne 14: "We also performed affinity selection with libraries containing up to 750 thousand members. In these selections, we decoded and validated several nanomolar binders for CAIX and inhibitors for FEN1" From what I understood, the hits were identified from a biased collection of 4,000 compounds and not from the 750k set.

-Experimental part: Could you explain the reason why the CAIX screening is run with a 1000 x excess of protein while in FEN1 a 10 folds excess of ligand is used.

-Supplementary: Page 25 & 26. Fig17 Difference in the selection parameters between the MS1 mass accuracy 5PPM in the text versus the 10PPM parameter selected in the interface window.

(Remarks on code availability)

Reviewer #3

(Remarks to the Author)

The manuscript of Boecker, Pomplun and co-workers describes the construction and affinity-based selection of "self-encoded" small molecule libraries (SELs) of sizes up to 500'000 compounds which are "encoded" and identified after selection through their fragment masses by LC MS-MS. To this aim the authors constructed 3 SELs: a 3-building block (BB) SEL1 of 2 amino acids and a capping carboxylic acid (500k compounds), a 3-BB SEL2 of a benzimidazole scaffold (216k compounds) and a 3-BB SEL2 including a Suzuki-Miyaura reaction step (31.8k compounds). The SELs were designed to fulfill the general Lipinski requirements and synthesized on solid phase. In a first step the MS methodology for detecting compounds was optimized, such that decent recall rate established. For achieving this, virtual databases of the potential library compounds were created, fragmentation patterns predicted and tested and compounds identified from LC-MS-MS runs showing the parental mass as well as at least one fragment observed by MS-MS. This procedure was necessary due to the redundancy of isobaric compounds in the SELs. The authors applied SIRIUS annotation workflow and automatized the procedure of spectra analysis (80'000 MS1 and MS2 scans) by setting-up a dedicated combinatorial mass decoding tool (termed "COMET"). This procedure allowed automated annotation and correct recall for the majority of library members. The authors then performed affinity-based selections in analogy to DNA-encoded libraries (DELs) on two targets, carbonic anhydrase IX, a generally used test target for DEL, and against flap endonuclease 1 (FEN1), a disease-related DNA-processing enzyme. The selections were performed against control beads without the target protein and the mass spectrometric analysis was performed such that hits of the controls were excluded. In the selection procedure the minimal amount of SELs was found to be 100 fmol/compound for reliable analysis. All SELs contained sulfonamide building blocks which are known to be submicromolar binders to CAIX and several hits were obtained from the 3 SELs, resynthesized and measure for affinity by BLI, leading to low-digit nanomolar hits. In the FEN1 selections, the SELs 1-3 did not give rise to hits, therefore a focused 4'000-member SEL4 was constructed containing N-hydroxy urea and trihydroxy phenol building blocks as anchors which helped the identification of high nanomolar hits against FEN1, which were resynthesized, tested by BLI and also docked to the protein structure. In summary, the authors have established a novel combinatorial library setup which is only intrinsically encoded by the compounds' masses and fragmentation, they have successfully devised a workflow for automated LC-MS-MS analysis, and shown the potential in the test system CAIX and for a focused smaller SEL for the novel target FEN1. While the lack of amplifiability and the MS resolution precludes the use of much larger SEL libraries than 1 million compounds, hampering its potential for de-novo discovery of low-affinity hits, it appears to me that this technology is very powerful and useful for smaller, focused libraries and hence can be complementary to DEL technology. The study is well executed and presented, I therefore support publication in Nat. Commun., after addressing the following points:

- The authors stress the "information density" of the SEL approach compared to DEL, yet this does not seem to me a decisive point, while commenting on the additional enzymatic steps necessary for DEL construction is reasonable.

- The authors claim SELs of a million compounds, yet have only tested a 500k SEL or combined SELs of ca. 750k compounds. Can they more clearly define the limits (e.g., with 100 fmol, potentially 10 fmol per compound)?

- The authors state in the introduction that the DNA-tag of DELs can substantially affect the selection process. This is not generally observed and may be true essentially for DNA-binding proteins. The author should rephrase this or substantiate their opinion with literature citations.

- The authors should discuss also potential solubility limitations of SEL.

In terms of hit identification:

- How much does the compound ionizability play a role? Does the enrichment correlate with ionizability? Did the authors investigate this in more detail? At least this point should be discussed.

- In DEL technology, enrichment can be determined by counting the DNA codes after selections. Here, does the workflow allow for a quantitative measure of enrichment of individual hits, or is it rather a yes or no response? How would you rank the obtained hits for follow-up?

- Does the LC and the MS lead to a bias of hits?

In terms of affinity-based selections:

- The authors state that the selections have been performed in duplicate, yet no respective data are given. Are the selections reproducible?

- Did the authors play with the stringency of the selections?

- How does the native (unselected) library decode, how many compounds could be identified in the 80k scans?

- the control selection was made against streptavidin-coated magnetic beads? You write about "unmodified" beads.

very minor points:

- in the reaction graphs of Fig. 2 and some SI Figures, Sn₂Cl is given instead of SnCl₂

- the "Supplementary LC-MS Data" is given as SI 10.xxx, it should be 12.xxx

In summary, I believe the presented work is a very interesting study and I support publication after the points raised above are adequately addressed.

(Remarks on code availability)

Version 1:

Reviewer comments:

Reviewer #1

(Remarks to the Author)

The reviewers have addressed all my comments, and the manuscript is in an excellent shape

>> SELs remain subject to a key limitation inherent to combinatorial libraries, namely, low scaffold diversity within individual libraries. However, this can be mitigated by the ability to rapidly generate distinct combinatorial libraries, thereby expanding the accessible chemical space. While complex polycyclic scaffolds may exceed the current capabilities of SEL decoding, many pharmaceutically relevant scaffolds could, in principle, be decoded using SIRIUS-COMET. Although generating very diverse libraries is more demanding than those based on a single scaffold, existing HTS libraries could be potentially pooled and enable rapid affinity selections, significantly streamlining the early phases of hit discovery.

I think this helps a lot with clarifying strength and limitations. Based on the preliminary results in the manuscript, I think this method is most useful when there is already a well-known lead that needs to be improved (e.g. CAIX system is well known to bind to sulfonamide) rather than initial lead discovery. I have a feeling that chance of success would be really low with synthetic libraries if you do not have any prior knowledge about well-known leads. I am not going to leave it up to authors whether they want to add such a discussion

(Remarks on code availability)

Reviewer #2

(Remarks to the Author)

The author has developed an intelligent deconvolution strategy that enables the identification of small-molecule protein binders within large compound libraries following an affinity selection screening assay. Unlike DNA-encoded libraries (DELs), these libraries are DNA-tag-free, which makes it possible to screen challenging targets such as DNA-binding proteins. The article is well written and allows a clear understanding of the approach. This work could increase the scope of ASMS in Hit-Identification.

(Remarks on code availability)

Reviewer #3

(Remarks to the Author)

The authors have convincingly addressed all the points raised by me, and as far as I could see, also the points of the other reviewers.

Clarifications and additional information (e.g., Figs. S16, S17) have been included and I now deem the manuscript suitable for publication in Nature Communications.

(Remarks on code availability)

Point-to-point reply to reviewer comments on **'Barcode-free hit discovery from massive libraries enabled by automated small molecule structure annotation'** by van der Nol et al.

Reviewer #1 (Remarks to the Author):

I am really impressed with the manuscript by van der Nol et al. For the computational aspects (EPIMETHEUS), I did not find them very innovative, as very similar in silico fragmentation approaches have been previously proposed. What really impressed me was the innovative application of these computational methods to high-throughput screening, with very detailed and unbiased explanation of bottlenecks (e.g. false positives) and strength of the approach. Additionally, the manuscript is very well written, and I believe the strengths of this manuscript and its innovativeness in application of in silico database search methods overcome its pitfalls. Please see my detailed comments below:

We thank reviewer 1 for their detailed review of our manuscript, for their positive response and for the valuable comments. We addressed all points as detailed here below, added additional explanations or text alterations to the manuscript where necessary and included additional statistical calculations as requested.

Major comments:

- *Has 4-sulfamoylbenzoic acid (individually) or 4-sulfamoylbenzoic acid containing molecule been previously reported to interact with the target? Why is there a need to test half a million molecules, if similar results could be achieved by just screening a small number of building blocks including 4-sulfamoylbenzoic acid?*

We selected the CAIX system precisely because it is well known to bind aromatic sulfonamides, including molecules such as 4-sulfamoylbenzoic acid. This prior knowledge makes it an ideal model system for validating our workflow, as we can evaluate selection outcomes even before extensive experimental binding validation. While a smaller library focused on sulfonamide-containing compounds might also yield hits, our goal was to challenge the full workflow, including affinity selection, enrichment, and decoding, under the most complex and demanding conditions, using our largest libraries. This comprehensive test allows us to confirm several critical aspects: (i) that the library behaves well under experimental conditions (e.g., solubility and compatibility), (ii) that potential contaminants or byproducts from bead elution do not interfere with decoding, and (iii) that sufficient quantities of individual compounds are retained to enable reliable MS-based detection and decoding.

Additionally, while the sulfonamide motif is a key binding element, the chemical context provided by neighboring building blocks can influence binding affinity and specificity – both positively and negatively. Screening larger, more diverse libraries increases the chance of identifying favorable combinations that may enhance binding beyond what is achievable by testing the core motif alone.

- *“Supplementary material by breaking either a single nonring bond or a pair of bonds within the same ring system.” This is exactly how Dereplicator+ fragmentation graphs are constructed (single nonring bonds are called bridges and pair of bonds within the same ring system are called 2-cuts). Some of the other existing fragmentation methods might also be using a similar concept.*

We agree with the reviewer that EPIMETHEUS is using ideas of previous combinatorial fragmenters. In fact, its problem definition is very similar to the one of FiD (Heinonen et al., 2006). The difference is that EPIMETHEUS reduces the search space of possible fragments by restricting that a fragment can only explain a peak if its molecular formula matches the one assigned by the fragmentation tree of SIRIUS. Note that SIRIUS' fragmentation trees assign MS2 peaks with molecular formulas and not structures. Furthermore, we agree that DEREPLICATOR+ (Mohimani et al., 2018) and also other previously reported methods (e.g. CFM-ID (Allen et al., 2014) or MetFrag (Wolf et al., 2010)) construct such fragmentation graphs in a very similar way. Therefore, we now also mention DEREPLICATOR+ and point out that non-ring bonds and pairs of bonds within a ring system are called *bridges* and *2-cuts* in a graph theoretical context.

- *“We filtered a comprehensive building block (BB) catalog (Fmoc-amino acids and carboxylic acids from Chemspace) by availability and price, narrowing it down to 1000 BBs per position” What was the number before narrowing down?*
- *With these we enumerated a virtual library with a billion members. Is it 1000*1000*1000? Please explain where “a billion” came from?*

The catalogue provided by Chemspace contained 1681 Fmoc protected amino acids (AA) and 6357 monofunctional carboxylic acids (CA). We narrowed it down to 1000 BBs for each functional group. The virtual library was enumerated using 1000*1000*1000 (CA-AA-AA) following the design of SEL1. We adjusted the text on page 5 line 30 accordingly.

- *we generated a half million-membered. Is half a million referring to 499720? If so, use the latter for clarity*
- *500k, 200k and 30k, respectively. Use exact numbers for clarity*

We have now added the exact numbers for each SEL in the manuscript and in the supplementary information.

- *Add a table and explain numbers 130, 62, 52, 67, 53, 10 and 60. Where each one is coming from. It's nice to have them all together.*

We expanded the explanation in the Figure caption on how we selected the building blocks used in SEL1-3 and the numbers for each position are shown in Fig 2d.

- *“we prepared defined subsets of 245 to 500 compounds for each scaffold and analyzed each subset via nanoLC-MS/MS”. Please clarify exactly how many from each set*

We have added a more explicit referral to the main text figure (3b) and the SI Figure 14, in which the exact number of building blocks and their structures are shown in detail.

- *“We implemented a filter, stipulating that only scans with an MS1 precursor mass matching a library compound’s mass and containing at least one predicted fragment peak in their MS2 would be selected for full annotation (Fig. 3f).” How specificity / sensitivity changes if 2, 3, etc. predicted fragments are required, instead of only 1 ?*

We now added Supplementary Table 2, which shows the number of hits and the total number of annotated compounds using 1, 2 or 3 matching fragments. Using 2 matching fragments generally increased the proportion of hits compared to false positives. While this approach retains most hits for SEL 1, the number of hits decreases substantially for SEL 2 and SEL 3. Determining the most suitable COMET setting is therefore dependent on the library design and the expected fragmentation pattern.

“74 different hits contained this substructure” How many hits did you get in total ? How many would you see by random ? What is the statistical significance ? (e.g. using fisher p-value)

- *“30 and 47 individual compounds, for SEL 2 and 3, respectively” Out of how many ? How many would you see by random ? What is the statistical significance ?*

In the affinity selections we identified 74 sulfonamides out of 228 total annotated compounds (SEL1), 30 out of 75 total (SEL2) and 47 out of 51 total (SEL3). These numbers are now added to the main text and to Figure 4a.

The statistical significance of finding the observed number of hits with a sulfonamide substructure compared to what would be expected from randomly selecting molecules from our libraries was calculated using Fisher’s exact test. Let N be the size of the screening library, K the number of structures in that library showing the sulfonamide moiety, and n the observed number of molecules after the ASMS workflow. Now, let X be a random variable describing the number of observed hits with such a sulfonamide substructure. X is hypergeometrical distributed and therefore, it is:

$$P(X = k) = \frac{\binom{K}{k} \binom{N-K}{n-k}}{\binom{N}{n}}$$

For computing the p-value of observing k hits with a sulfonamide substructure, we have to compute the probability $P(X \geq k)$ of observing k or more hits showing the sulfonamide substructure under the assumption of the null hypothesis (random selection of the molecules). Those p-values are computed for every library with the values for k being 74, 30, and 47. The computation was done via the fisher.test function in R-studio.

The corresponding calculations, contingency tables, and results are shown in Supplementary Figure 18 and referred to in the main text. The p values are $p = 8.362941e-97$ (SEL1), $p = 1.00912e-31$ (SEL2) and $p = 3.45304e-77$ (SEL3) showing a strong statistical significance.

- “ 2.2×10^3 (74/228:74/499,720), 2.9×10^3 (30/75:30/216,008) and 6×10^2 (47/51:47/31,800)” I am a little confused, so #found binders = #total binders in all cases? What does it mean? can you explain?
- “74/228:74/499,720” What is 228?
- “[(# found binders)/(# compounds after selection)]:[(# total binders)/(# total library)].” Can you improve the presentation of this formula, aesthetically?

We have now improved the presentation of the formula in the text (we will also make sure it is clearly readable when working on the final version of the article).

With this formula, we calculate what generally referred to as the *maximum enrichment*. In a library of 499,720 compounds, there is an unknown number of true binders. Determining the exact number of binders would require testing each compound individually, which is not feasible at this scale. However, based on prior knowledge that aromatic sulfonamides can mediate binding to our target, we make a simplifying assumption: all selected hit structures that contain this motif are considered binders. While this may underestimate or overestimate the true number—since other motifs might also contribute to binding, or some sulfonamide-containing compounds might not actually bind, it provides a consistent and reasonable baseline. Since we cannot determine the true number of binders in the entire library, we calculate the *maximum enrichment* by assuming that all observed hits represent the full set of binders. Enrichment is then defined as the ratio between:

The fraction of binders (i.e., sulfonamide-containing hits) among the selected/identified compounds, and fraction of those same structures within the entire library. In the specific example 228 are all annotated compounds after selection. 74 of which are sulfonamides.

This provides an upper bound on enrichment, hence the term *maximum enrichment* and allows us to evaluate the performance of the selection process, even in the absence of exhaustive experimental validation. The concept of *maximum enrichment* is commonly used in affinity selection workflows (see e.g., <https://www.nature.com/articles/s41467-020-16920-3>).

- “90 vs. 150” 150 is not consistent with any of the numbers you previously reported: 2.2×10^3 (74/228:74/499,720), 2.9×10^3 (30/75:30/216,008) and 6×10^2 (47/51:47/31,800)

We apologize for the confusion. When mixing all libraries together and performing the experiment, we identified 90 sulfonamide hits. From the individual library selections we had identified $74 + 30 + 47 = 151$ hits. We adjusted the text to make it more clear and adjust the mistake.

- “we have tested the decoding on three different library architectures.” Three? or four?

We describe the use of 3 different library architectures in terms of connectivity between building blocks. SEL1 and SEL4 have the same library architecture but use a different number and type of building blocks.

- Even after application of EPIMETHEUS threshold, false discovery is still high. I acknowledge going from half a million to only a few hundred FPs is still an impressive job. However, The authors should expand (at least in discussion) how these false positives will impact downstream analysis.

We added this discussion point, acknowledging that with the current filtering, manual analysis of the final set of candidates is still a requirement. We also indicated that further research will be needed to refine the filtering and improve structure ranking, in order to enable the practical screening of large libraries.

- *It is well known in drug discovery that these sort of combinatorial libraries (Sel 1 – Sel 4) are less diverse and less interesting than more diverse alternatives. I was expecting a discussion of this, along with potential bottlenecks of applying the proposed method to more diverse libraries (it is just the issue of cost of library?)*

We added a discussion paragraph about diversity limitations and potential future directions.

Reviewer #2 (Remarks to the Author):

The author present a new approach allowing the screening of large collections of small molecules by affinity selection without specific library encoding. Hit compounds are characterized by a MS1/MS2 fragment mass encoding. The strategy has been used and described for large peptide libraries but not for small molecules. The article is very well written and pleasant to read. Despite progresses described in the annotation, the percentage of compounds correctly annotated remains quite low to my perception and should be presented as such. The article is well documented and detailed information given to facilitate reproduction of the work.

We thank reviewer 2 for taking the time for reviewing our manuscript, for their positive response and for the valuable comments. We addressed all points as detailed here below and added additional explanations or alterations to the manuscript where necessary.

Comments to improve the quality of the article:

- *Abstract: “screening barcode-free libraries of this scale is unprecedented” Not really true. Larger libraries of peptides have been screened in AS-MS and multi-million small molecule collections screened using pooling strategies.*

We adjusted the statement to ‘screening barcode-free libraries of this scale *all at once* is unprecedented’

- *Page 7, line 41: How do you check that compounds are annotated correctly?*

For the miniSELS, we manually reviewed each annotated compound. To streamline this process, we first performed a SIRIUS run using only the exact miniSEL database and manually evaluated the resulting annotations. This provided a reliable reference point, increasing our confidence when analyzing the same sample against the larger, full SEL dataset.

- *Figure 3: “sublibrary with less than 500 unique compounds” Unique compounds refers usually to single compounds. I understand that those compounds are produced as mix of compounds.*

We adjusted to ‘For each library scaffold a small combinatorial sublibrary with 245 or 500 different compounds was synthesized’ to avoid misunderstanding.

- *Fig3b : The table presented to illustrate the COMET annotation should be comparable to the Automated SIRIUS 6 analysis table. Data should be presented in the same order, that is, Annotated structures, true member recall and Correct annotations (missing column on purpose?). The correct annotation rate is only 16% for library SEL1 and 41 % for SEL2, 64% for SEL 3. Despite the improvement obtained as compared to the automated SIRIUS 6 analysis, the values obtained for SEL 1 & 2 are far from being a “high correct annotation rate”. I will recommend to use “Moderate correct annotation rate”*

We adjusted Figure 3 as requested.

- *Page 11, line 24: “which were substantially enriched over background” Could you be more specific and state the enrichment cut-off ratio used to separate hits from background.*

As an initial data filtering step we do a background subtraction using a 3-fold max intensity ratio threshold in combination with a MS m/z accuracy of 5ppm and RT accuracy of 10 seconds. These details are now added to the Supplementary Information section 4 and in Supplementary Fig 20.

When immobilizing proteins via His the general background signals are higher as compared to the biotin immobilization. For the FEN1 examples the hit we also used a cut-off of 3-fold max intensity ratio over control. Compounds **6** and **7** have a 3.9 and 3.8 fold intensity ratio over background, respectively. This detail has now been added to page 15, line 25. In the CAIX examples, due to the better screening conditions, in most cases no peak corresponding to the hit compound was observed at all in the control selection.

- *Page 13, line 14: “We also performed affinity selection with libraries containing up to 750 thousand members. In these selections, we decoded and validated several nanomolar binders for CAIX and inhibitors for FEN1” From what I understood, the hits were identified from a biased collection of 4,000 compounds and not from the 750k set.*

We apologize for the confusion. The text is now changed to the following sentence: “ We also performed affinity selection with libraries containing up to 750 thousand members against CAIX and with 4 thousand members against FEN1”

- *Experimental part: Could you explain the reason why the CAIX screening is run with a 1000 x excess of protein while in FEN1 a 10 folds excess of ligand is used.*

Thanking you for spotting this mistake. The selection against FEN1 was performed using 1 pmol/member (and not 1 nmol/member as mentioned in the method section). We have now corrected the mistake.

In general we do use excess of protein to prevent competition among library members. For the CAIX selection, a 1000-fold excess of protein relative to each library member was used. For the much smaller library used for the FEN1 selection, we increased the library concentration to 1 pmol/member, still corresponding to a 100 fold excess of protein relative to each library member.

- *Supplementary: Page 25 & 26. Fig17 Difference in the selection parameters between the MS1 mass accuracy 5PPM in the text versus the 10PPM parameter selected in the interface window.*

We adjusted Supplementary Fig 20. The 5 ppm mass accuracy is now accurately depicted in the figure.

Reviewer #3 (Remarks to the Author):

The manuscript of Boecker, Pomplun and co-workers describes the construction and affinity-based selection of "self-encoded" small molecule libraries (SELs) of sizes up to 500'000 compounds which are "encoded" and identified after selection through their fragment masses by LC MS-MS. To this aim the authors constructed 3 SELs: a 3-building block (BB) SEL1 of 2 amino acids and a capping carboxylic acid (500k compounds), a 3-BB SEL2 of a benzimidazole scaffold (216k compounds) and a 3-BB SEL2 including a Suzuki-Miyaura reaction step (31.8k compounds). The SELs were designed to fulfill the general Lipinski requirements and synthesized on solid phase. In a first step the MS methodology for detecting compounds was optimized, such that decent recall rate established. For achieving this, virtual databases of the potential library compounds were created, fragmentation patterns predicted and tested and compounds identified from LC-MS-MS runs showing the parental mass as well as at least one fragment observed by MS-MS. This procedure was necessary due to the redundancy of isobaric compounds in the SELs. The authors applied SIRIUS annotation workflow and automatized the procedure of spectra analysis (80'000 MS1 and MS2 scans) by setting-up a dedicated combinatorial mass decoding tool (termed "COMET"). This procedure allowed automated annotation and correct recall for the majority of library members. The authors then performed affinity-based selections in analogy to DNA-encoded libraries (DELs) on two targets, carbonic anhydrase IX, a generally used test target for DEL, and against flap endonuclease 1 (FEN1), a disease-related DNA-processing enzyme. The selections were performed against control beads without the target protein and the mass spectrometric analysis was performed such that hits of the controls were excluded. In the selection procedure the minimal amount of SELs was found to be 100 fmol/compound for reliable analysis. All SELs contained sulfonamide building blocks which are known to be submicromolar binders to CAIX and several hits were obtained from the 3 SELs, resynthesized and measure for affinity by BLI, leading to low-digit nanomolar hits. In the FEN1 selections, the SELs 1-3 did not give rise to hits, therefore a focused 4'000-member SEL4 was constructed containing N-hydroxy urea and trihydroxy phenol building blocks as anchors which helped the identification of high nanomolar hits against FEN1, which were synthesized, tested by BLI and also docked to the protein structure. In summary, the authors have established a novel combinatorial library setup which is only intrinsically encoded by the compounds' masses and fragmentation, they have successfully devised a workflow for automated LC-MS-MS analysis, and shown the potential in the test system CAIX and for a focused smaller SEL for the novel target FEN1. While the lack of amplifiability and the MS resolution precludes the use of much larger SEL libraries than 1 million compounds, hampering its potential for de-novo discovery of low-affinity hits, it appears to me that this technology is very powerful and useful for smaller, focused libraries and hence can be complementary to DEL technology. The study is well executed and presented, I therefore support publication in Nat. Commun., after addressing the following points:

We thank reviewer 3 for their time to review the manuscript, their positive response and their suggested improvements. We addressed all points as detailed here below, added additional explanations or text alterations to the manuscript where necessary and included additional data analysis as requested.

- *The authors stress the "information density" of the SEL approach compared to DEL, yet this does not seem to me a decisive point, while commenting on the additional enzymatic steps necessary for DEL construction is reasonable.*

We removed the concept of 'information density' from the introduction paragraphs. We would, however, like to keep it in one of the discussion paragraphs. The maximum information density of SELs even if not the most important, is still a valid point: it means the molecules are screened in their completely unmodified form and do therefore not have any influence of additional features attached to them.

- *The authors claim SELs of a million compounds, yet have only tested a 500k SEL or combined SELs of ca. 750k compounds. Can they more clearly define the limits (e.g., with 100 fmol, potentially 10 fmol per compound)?*

We indeed have tested only up to 750k compounds as the biggest library size. We did revise the statement in the abstract to 'over half a million'.

In the chapter about the CAIX screenings (end of page 11) we do show our investigation towards the limits of selection conditions. We tested selections with 1 pmol, 100 fmol and 10 fmol per compound. No substantial difference was observed between 1 pmol and 100 fmol, but selection with 10 fmol had significantly decreased hit ID rates. Based on these experiments we made two conclusions. Screening at 100-1000 fmol per members are ideal. And, given that a library of 500.000 members can be screened at 1000 fmol without detectable solubility problems, a library of 5 million members screened at 100 fmol/member could in principle also be possible.

- *The authors state in the introduction that the DNA-tag of DELs can substantially affect the selection process. This is not generally observed and may be true essentially for DNA-binding proteins. The author should rephrase this or substantiate their opinion with literature citations.*

We rephrased the statement to: "*which can potentially affect the selection process by restricting the binding pose diversity of each library member, or by interacting with the target and leading to false positives or false negatives.*" We also added a reference to the recent article by Raphael Franzini, showing how the DNA tag can indeed influence selection outcomes.

(<https://pubs.rsc.org/en/content/articlelanding/2025/sc/d5sc00844a>)

- *The authors should discuss also potential solubility limitations of SEL.*

As discussed above, we performed affinity selections with the 500.000 membered library at different concentrations (see end of page 12). Working at 1 pmol/member (with selection in 1 mL resulting in a

1 nM per member and 500 uM overall library concentration) was feasible and did not result in detectable solubility problems. We added a statement to that paragraph that even larger library sizes will likely start to cause solubility problems.

- *In terms of hit identification: How much does the compound ionizability play a role? Does the enrichment correlate with ionizability? Did the authors investigate this in more detail? At least this point should be discussed.*

Different compounds do vary in their ionizability. The challenge is well recognized in the field of metabolomics and proteomics. However, among all the many hits selected in this study, no bias toward typical functional groups with enhanced ionizability was observed (see for example SI Fig 20, page 44). We added this observation to the first paragraph about the CAIX selection. Based on these results we are confident to state that actual binding to the target is a way more important determinant for a compound to show up as a hit compound after washing, eluting, computational filtering and analysis. We do observe that ionizable fragments give stronger peaks in the MS2 spectra, but also this behavior does not influence the overall decodability of the compounds.

In DEL technology, enrichment can be determined by counting the DNA codes after selections. Here, does the workflow allow for a quantitative measure of enrichment of individual hits, or is it rather a yes or no response? How would you rank the obtained hits for follow-up?

As noted by this reviewer and discussed before, different compounds do have different ionizability, even if these do not bias the selection outcomes. A direct comparison between mass spec signal intensities therefor does not give direct indication of which compound had stronger binding and better enrichment. Under the conditions used in this study, we therefore do indeed use the readout as a yes/no response. Selection of hits for follow up can be mainly based on enriched building blocks or building block groups.

- *Does the LC and the MS lead to a bias of hits?*

In terms of LC (and sample prep) there might be loss of certain overly polar compounds. These compounds might be washed away during the desalting step in stage tipping or might elute in the first minute of the LC run, when no MS is recorded. Examples of such compounds might be doubly positively charged molecules like X-Lys-Lys or X-Arg-Arg. As such very polar compounds arguably would not have very good drug like properties their loss can be considered a very minor limitation.

In terms of MS, as discussed above, we did not observe any specific bias. Hits contain positive, negative, and neutral building blocks and also in the miniSEL test libraries we did not observe any specific functional groups that lead to problems in detection.

- *In terms of affinity-based selections: The authors state that the selections have been performed in duplicate, yet no respective data are given. Are the selections reproducible?*

We now added Supplementary Figure 16 where we show in a Venn diagram how many hits were identified on both duplicates. The diagrams show a moderate degree of overlap of annotated hit compounds. It is important to note that the actual reproducibility of the selection per se is much higher

than the overlap of ultimately annotated hit compounds. The fact that the overlap is not full can depend on a number of factors and might mainly have to do with the mass spectrometry analysis and the structure annotation. A compound can be present after both selections, but the mass spec data from one run might not be of high enough quality to enable structure annotation. We provided examples in Supplementary Fig. 17 where we show a clear enrichment in MS1 data for several hits that were annotated in only one of the two duplicates of the selection. Overall these data indicate that the selection have good reproducibility, but to enable decoding and identification of most possible hits, running replicates can be very important.

- *Did the authors play with the stringency of the selections?*

We did not vary the stringency of the selections (beside the variations of the library concentrations discussed above). All validated hits for CAIX had low nanomolar binding affinity, so in that case it did not seem of importance testing more stringent conditions. For FEN1 we did identify only 2 hits, and any more stringent conditions would have potentially resulted in losing all hits. We do agree that in some cases selection stringency might be of importance and we will take this along in follow up studies.

- *How does the native (unselected) library decode, how many compounds could be identified in the 80k scans?*

The full library of 500,000 compounds was never injected into the nanoLC-MS/MS system. Injecting such a large library at concentrations suitable for detection and decoding would be impractical, posing a high risk of instrument damage and column clogging. Moreover, the resulting sample density would lead to significant ion suppression. The 80,000 scans referenced in the text correspond to the experiments involving the injection of miniSEs 1–3 into the instrument.

- *the control selection was made against streptavidin-coated magnetic beads? You write about "unmodified" beads.*

For clarity we adjusted “unfunctionalized magnetic beads” to “streptavidin-coated magnetic beads”

very minor points:

- *in the reaction graphs of Fig. 2 and some SI Figures, Sn2Cl is given instead of SnCl2*
- *the "Supplementray LC-MS Data" is given as SI 10.xxx, it should be 12.xxx*

We have corrected both mistakes.

Point-to-point reply to reviewer comments on **'Barcode-free hit discovery from massive libraries enabled by automated small molecule structure annotation'** by van der Nol et al.

REVIEWERS' COMMENTS

Reviewer #1 (Remarks to the Author):

The reviewers have addressed all my comments, and the manuscript is in an excellent shape

>> SELs remain subject to a key limitation inherent to combinatorial libraries, namely, low scaffold diversity within individual libraries. However, this can be mitigated by the ability to rapidly generate distinct combinatorial libraries, thereby expanding the accessible chemical space. While complex polycyclic scaffolds may exceed the current capabilities of SEL decoding, many pharmaceutically relevant scaffolds could, in principle, be decoded using SIRIUS-COMET. Although generating very diverse libraries is more demanding than those based on a single scaffold, existing HTS libraries could be potentially pooled and enable rapid affinity selections, significantly streamlining the early phases of hit discovery.

I think this helps a lot with clarifying strength and limitations. Based on the preliminary results in the manuscript, I think this method is most useful when there is already a well-known lead that needs to be improved (e.g. CAIX system is well known to bind to sulfonamide) rather than initial lead discovery. I have a feeling that chance of success would be really low with synthetic libraries if you do not have any prior knowledge about well-known leads. I am not going to leave it up to authors whether they want to add such a discussion

Reply: Thank you for the support for our manuscript. We did not add an additional discussion point about needing specific building blocks to have success in binder identification. There is no clear reason to believe, that such big libraries cannot result in early de novo hit identification and for the CAIX example we already indicated the importance of that specific functional group.

Reviewer #2 (Remarks to the Author):

The author has developed an intelligent deconvolution strategy that enables the identification of small-molecule protein binders within large compound libraries following an affinity selection screening assay. Unlike DNA-encoded libraries (DELs), these libraries are DNA-tag-free, which makes it possible to screen challenging targets such as DNA-binding proteins. The article is well written and allows a clear understanding of the approach. This work could increase the scope of ASMS in Hit-Identification.

Reply: Thank you for the support for our manuscript.

Universiteit
Leiden
The Netherlands

LACDR

Sebastian J. Pomplun

Associate Professor

Phone: +31(0)71 527 4651

Email: s.j.pomplun@lacdr.leidenuniv.nl

Website: pomplunlab.com/

Reviewer #3 (Remarks to the Author):

The authors have convincingly addressed all the points raised by me, and as far as I could see, also the points of the other reviewers.

Clarifications and additional information (e.g., Figs. S16, S17) have been included and I now deem the manuscript suitable for publication in Nature Communications.

Reply: Thank you for the support for our manuscript.